# CRISPR/Cas Technology Revolutionizes Crop Breeding

**DOI:** 10.3390/plants12173119

**Published:** 2023-08-30

**Authors:** Qiaoling Tang, Xujing Wang, Xi Jin, Jun Peng, Haiwen Zhang, Youhua Wang

**Affiliations:** 1National Nanfan Research Institute (Sanya), Chinese Academy of Agricultural Sciences, Sanya 572024, China; tangqiaoling@caas.cn; 2Biotechnology Research Institute, Chinese Academy of Agricultural Sciences, Beijing 100081, China; wangxujing@caas.cn; 3Hebei Technology Innovation Center for Green Management of Soi-Borne Diseases, Baoding University, Baoding 071000, China; jinxi2007@126.com

**Keywords:** CRISPR/Cas technology, crop, germplasm, breeding technology

## Abstract

Crop breeding is an important global strategy to meet sustainable food demand. CRISPR/Cas is a most promising gene-editing technology for rapid and precise generation of novel germplasm and promoting the development of a series of new breeding techniques, which will certainly lead to the transformation of agricultural innovation. In this review, we summarize recent advances of CRISPR/Cas technology in gene function analyses and the generation of new germplasms with increased yield, improved product quality, and enhanced resistance to biotic and abiotic stress. We highlight their applications and breakthroughs in agriculture, including crop de novo domestication, decoupling the gene pleiotropy tradeoff, crop hybrid seed conventional production, hybrid rice asexual reproduction, and double haploid breeding; the continuous development and application of these technologies will undoubtedly usher in a new era for crop breeding. Moreover, the challenges and development of CRISPR/Cas technology in crops are also discussed.

## 1. Introduction

In the future, agricultural production faces major challenges from a rapidly increasing human population and severe environmental stresses. Crop yield is a complex quantitative trait governed by many genes and environment factors, and some key genes and agronomic traits have gradually weakened or been lost during crop domestication [1]. Traditional breeding methods, such as crossing breeding and mutation breeding, have achieved increasing crop yield, but they also have many limitations in breeding superior varieties due to the lack of valuable natural germplasms, the obstacles of undesired genome incorporation or linkage drag, and their time consuming and laborious screening process [2]. Compared with traditional methods, biotechnologies including the use of transgene, gene editing, double haploid technique, and synthetic apomixis provide new opportunities for crop breeding [2,3,4,5,6]. Transgenic technology by ectopically expressing specific genes overcomes the limitations associated with conventional techniques and has played an important role in the genetic improvement of crop yield, quality, and resilience towards biotic and abiotic stresses; however, its use has been seriously restricted due to the time-consuming process, risk assessment, and regulatory process [7]. As an important technique, doubled haploid technology can effectively accelerate the development of new crop varieties and shorten the breeding period by several years through the directly generating homozygous plants, which have been widely used in multiple major crops such as maize, wheat, rice, and soybean. The combination of double haploid breeding with traditional breeding methods, molecular marker-assisted selection, and especially gene editing technology will greatly improve the efficiency of crop breeding [8,9].

Mutations are the basis of the creation of new genetic resources and crop breeding. Plant mutagenesis techniques include physical and chemical mutagens, RNA interference, transcription activator-like effector nucleases (TALENs), zinc-finger nucleases (ZFNs), and clustered regularly interspaced short palindromic repeats (CRISPR)-associated (Cas) (CRISPR/Cas). The advent of the CRISPR/Cas system provides a promising platform for genome editing in a site-specific manner and initiates a new era in which the researchers can rapidly and precisely create novel germplasms by manipulating key genes responsible for specific agronomic traits [10]. Based on the initially engineered CRISPR/Cas9, diverse CRISPR tools have been developed for precise and effective genome editing across all layers of the central dogma processes [11]. The expanding CRISPR toolbox enables gene knockout, gene deletion, base editing, gene insertion or replacement, targeted random mutagenesis, epigenetic modulation, transcriptional regulation, and RNA editing [12,13,14]. The CRISPR/Cas system iterative upgrades have provided unprecedented opportunities for gene function analyses and the creation of desirable germplasms in different crops, which will lead to the third agricultural green revolution [15,16]. The combination of CRISPR/Cas technology with traditional breeding methods, molecular marker-assisted selection, double haploid technique, male or female sterility, and asexual reproduction will greatly promote crop breeding, In this review, we mainly summarize the application of CRISPR/Cas technology in gene function analysis, the generation of new germplasms, crop *de novo* domestication, decoupling the tradeoff effect, conventional hybrid seed production, asexual reproduction, and double haploid breeding [17].

## 2. Exploring Gene Functions and Creating Desired Germplasms

CRISPR/Cas technology has unparalleled advantages in characterizing gene functions and faster generation of valuable crop germplasm resources [18]. CRISPR/Cas systems enable the researchers to manipulate genes via gene knockout, gene knock in, gene replacement, gene base editing, gene regulation, and epigenome editing, which opens an era for studying gene function in different crops [19]. For example, *KRN2* has been well characterized as a convergent selected gene for the regulation of grain number in both maize and rice through CRISPR/Cas technology. Knockout of *KRN2* significantly increased their grain yields without affecting other agronomic traits, providing a feasible strategy for the generation of new germplasm and crop *de novo* domestication [20]. In maize, upright plant architecture is a practical choice for high-density planting and high yield [21]. CRISPR/Cas9 editing *ZmRAVL1*, a positive regulator of leaf angle, engineered an upright plant architecture with increased yield under high planting densities, providing an option to develop density-tolerant high-yield cultivars [21,22]. Using CRISPR/Cas technology, the *DUYAO* was identified as the candidate gene of *RHS12* locus controlling male gamete sterility in indica-japonica inter-subspecific hybrid rice, which clarified the genetic basis of reproductive isolation and provided a strategy for molecular designs of hybrid rice breeding [23].

Currently, CRISPR/Cas technology has been widely used in the improvement of crop traits, including biotic and abiotic resistance, plant development and morphology, crop yield, as well as grain nutrition and quality [19]. Recently, specialty corns including sweet, waxy, and baby corns have a growing consumer demand and, therefore, to meet this need, supersweet and waxy corns and aromatic corns were created by simultaneously editing *ZmBADH2a*/*b*, *SH2*, and *WX*, respectively [24,25], meeting consumer demand for the t aromatic corns with an appetizing fragrance or sweet and waxy corns. In the USA, CRISPR-waxy corn hybrids with higher-yield and superior agronomic performance to conventional lines were generated and pre-commercialized [26]. Southern leaf blight is a major foliar disease which causes significant yield losses worldwide. *ChSK1*-edited maize showed durable resistance to southern leaf blight, providing the potential strategy for engineering durable disease resistance maize germplasm [27]. Moreover, a gene discovery pipeline BREEDIT, combining multiplex genome editing of whole gene families with crossing schemes, has been used to identify valuable alleles for complex quantitative maize trait improvements [28], providing a feasible tool to identify key genes and cultivate desired maize lines.

In rice, many new germplasms with higher eating and nutritional quality (e.g., low amylose content; low glutelin content and grain aroma); improved agronomic traits (e.g., grain shape; tiller number and grain yield), or stress resistance were generated by editing *FLO2*; *Wx*; *OsBADH2*; *GS3*, *TGW6*; *SSII-2*; *SSII-3*; *OsPLDα1*; *OsAAP3*; *OsAAP4*; *OsAAP5*; *OsSPMS1*; *OsRR30*; *Xa13*; *Bsr-d1*; *Pi21*; *ERF922*; *OsSWEET1b*; *OsWRKY63*; and *JMJ710*, respectively [29,30,31,32,33,34,35,36,37,38,39,40,41,42]. In wheat, multiple new germplasms with increased grain yield, improved grain quality, and enhanced stress resistance were created by editing *FT-D1*; *Gli-γ1-1D*; *Gli-γ2-1B*; *pinb*, *waxy*, *ppo* and *psy*, respectively [43,44,45]. These studies provide potential strategies to develop new germplasms with high value specialty foods, increased yield, and enhanced biotic and abiotic stress tolerance in rice and wheat.

Soybean is an important oil-bearing crop, whose oils are primarily composed of polyunsaturated fatty acids, monounsaturated fatty acids, and saturated fatty acids. Comparatively, the high content of monounsaturated fatty acids in vegetable oils is beneficial for human health and food shelf-life [46]. Recently, an elevated monounsaturated fatty acids soybean germplasm was created by knocking-out *GmPDCTs*, providing a feasible strategy for the improvement of soybean nutrition and shelf stability [46]. Soybean root rot disease is a most destructive disease majorly caused by *Phytophthora sojae* (*P. sojae*) that endangers the entire growth cycle of soybean and seriously affects its yield and quality [47]. Recently, an excellent germplasm with high resistance to several *P. sojae* strains was engineered by editing *GmTAP1* [47]. Moreover, key genes associated with yellow-green variegation of leaf [48], plant architecture [49], and fatty acid anabolic metabolism [50] have been well characterized in soybean.

## 3. Ushering in a New Era of Crop *De Novo* Domestication

For a long time, crop domestication has mainly focused on selecting desirable traits related to yield, leading to the gradual loss of potentially useful traits in wild species such as pest and disease resistance, abiotic stress tolerance, and nutritional quality. Traditional wild crop domestication methods are still an option to create new germplasms, but they are time-consuming, labor-intensive, and difficult to aggregate multiple traits [51,52]. In 2017, a strategy for wild species *de novo* domestication was proposed using genome editing techniques [52]. In particular, CRISPR/Cas technology has achieved rapid wild crop *de novo* domestication by simultaneously editing key genes controlling agronomic traits, showing attractive prospects for effectively developing elite varieties [51,52,53,54,55,56,57].

In tomato, alleles conferring desirable traits were discovered in wild *Solanum pimpinellifolium*. These allelic changes were engineered by editing six important loci essential for yield and nutritional value in modern tomatoes. The engineered lines had significant increases in fruit size, fruit number, and fruit lycopene [58]. Novel germplasms with enhanced productivity were created in the orphan Solanaceae crop ‘groundcherry’ (*Physalis pruinosa*) by editing orthologues of modern tomato genes controlling plant architecture and yield-related traits, realizing the rapid creation of elite genomic resources in distantly related orphan crops [59]. Stress-tolerant wild-tomato species with desirable traits were engineered using a multiplex CRISPR/Cas9 technology, which had domesticated phenotypes and retained parental stress-tolerant traits [60]. Moreover, a ‘two-in-one’ strategy for stress-tolerant and multi-scenario cultivation breeding in tomatoes was devised through combining wild species *de novo* domestication with CRISPR/Cas, generating male-sterility in modern cultivars [61]. Recently, rapid *de novo* domestication of wild allotetraploid rice was realized by editing *O. alta* homologs of the genes controlling various agronomic traits in diploid rice, providing an effective way to breed new varieties aggregating desired traits via wild rice *de novo* domestication [51,62]. These studies demonstrate that CRISPR/Cas technology is a powerful tool for wild species *de novo* domestication to create new desirable varieties, ushering in a new era of crop breeding to utilize wild species genetic diversity in major and orphan crops.

## 4. Breaking Breeding Bottlenecks of Tradeoff Effects

Due to linkage drags or gene pleiotropy, crop breeding is often subject to complex tradeoffs between traits, such as high yield and stress/disease resistance, yield and nutritional quality, and yield and plant architecture. In particular, the tradeoff effects caused by gene pleiotropy have become the bottleneck of multi-traits pyramiding breeding [63,64,65,66]. Recently, CRISPR/Cas-mediated editing of cis-regulatory regions was used in different crops to generate novel beneficial alleles with improved stress resistance, yield, and quality [67,68]. Unlike editing a coding region, editing a cis-regulatory region can fine-tune the expression level or profile of the target gene without disrupting its function, thereby optimizing the tradeoff effects of the pleiotropic gene [67,69]. Recently, various tradeoff effects have been subtly solved in rice, maize, and wheat using CRISPR/Cas technology.

In rice, OsSWEET14 has tradeoff effects on disease resistance with plant height, tiller number, and seed size, whose loss-of-function mutation confers rice with enhanced disease resistance, but leads to small seed and delayed growth [70,71]. Using CRISPR/Cas technology, a broad-spectrum resistant rice with normal tiller number and seed size was developed by editing the TALE-binding elements in *OsSWEET11* and *OsSWEET14* promoters [72]. *IPA1*, a pleiotropic gene regulating various agronomic traits and stress resistance, has tradeoff effects on rice yield-related traits such as panicle size and tiller number [73,74,75,76,77,78,79,80,81,82]. New rice germplasms with enhanced yield were created by deleting a key cis-regulatory region controlling *IPA1* expression pattern, which subtly decoupled its tradeoff effect on grains per panicle and tiller number [66,83]. *SLG7* is a key gene regulating grain slenderness and low chalkiness. By editing the AC II element-containing region in the promoter, CRISPR/Cas-edited *SLG7* alleles with increased expression levels exhibited better appearance quality without affecting yield and eating quality [84]. Similarly, in hybrid rice, editing the regulatory regions of *HEI10* led to an altered expression level and genetic recombination, which may be used for developing elite varieties [85]. Recently, a CRISPR/Cas12a promoter editing (CAPE) system has been developed to improve rice agronomic traits by editing specific gene promoters. A high yield rice with better lodging resistance than Green Revolution *OsSD1* mutant was generated by editing the *OsD18* promoter [86]. Moreover, a high-efficiency prime-editing system was used to create resistant alleles with broad-spectrum resistance by knocking-in TAL effector binding elements from *OsSWEET14* into the promoter of dysfunctional *xa23* [87]. These research examples provide alternative strategies for the creation of quantitative variations of agronomic traits.

In maize, although the key components of the CLAVATA-WUSCHEL signal pathway impact yield formation due to their tradeoff effects on ear meristem activity and ear development, their weak alleles show few yield effects [88,89,90,91,92,93]. Recently, elite weak alleles with increased grains per ear and maize yield were created by editing the key regulatory regions of *ZmCLE7* and *ZmFCP1* [94]. Since the CLAVATA-WUSCHEL signal pathway that coordinates stem cell proliferation with differentiation is conserved in diverse higher plant species, this study provides an alternative idea to improve yield traits of other crops.

In wheat, *Mildew resistance locus O* (*MLO*), a pleiotropic susceptibility gene, has tradeoff effects on disease resistance and yield-related traits [44,95,96]. Using CRISPR/Cas technology, a *mlo* resistance allele (*Tamlo*-R32) with normal growth and yield was engineered in multiple varieties by deleting a large fragment in the *MLO*-B1 locus, which led to ectopic activation of *TaTMT3B* and thereby rescued growth and yield penalties caused by *MLO* disruption [44]. These studies provide effective strategies for developing high-yielding crop varieties with stress resistance by decoupling the tradeoff on different traits caused by gene pleiotropy.

## 5. Accelerating Conventional Production of Crop Hybrid Seed

Heterosis is a breakthrough in crop breeding which has greatly improved crop yield. However, since offspring cannot maintain their heterosis due to genetic separation of traits, it is a time-consuming, laborious, and costly process to produce hybrid seeds every year [97,98]. The wide use of male-sterile lines has greatly enhanced hybrid seed yield and quality in crop breeding. Recently, CRISPR/Cas technology has shown its unique advantages to unravel the mechanism of male sterility and develop male-sterile lines [99]. Many male-sterile-related genes have been well characterized in maize [100], rice [101,102,103,104,105], wheat [106], and soybean [107], and improved the understanding of the molecular mechanisms that control male sterility in crops. Importantly, increasing CRISPR-edited male-sterile lines have been generated in maize by knocking out *ZmMS26* or *ms8* [108,109], rice by editing *TMS5*, *OsOPR7*, or *CYP703A3* [99,110,111,112,113,114,115], wheat by targeting *TaNP1*, *Ms1*, or *Ms45* [116,117,118], foxtail millet [119], and soybean [120]. Thus, the CRISPR/Cas technology has provided a powerful tool for the generation of male-sterile lines and will greatly promote commercial hybrid seed production in different crops.

In the production of hybrid rice seeds, the restorer lines must be removed before seed harvesting to avoid contaminating undesired inbred line seeds, which results in the labor-intensive and costly hybrid seed production. Thus, the use of female-sterile lines as pollen donors might be an effective strategy to solve this problem [98]. Unlike male-sterility, thermo-sensitive female sterility has been rarely reported due to the lack of desired germplasms, but it is very important for crop hybrid seed production via full mechanization [121]. Using CRISPR/Cas technology, the first thermo-sensitive female sterility gene, *AGO7/TFS1*, was identified to engineer a female sterility line without causing defects in vegetative or male reproductive development. As a restorer line, its field trial showed a high seed-setting rate of hybrid panicles, paving a new path for fully mechanized hybrid seed production like conventional rice [98,121].

## 6. Promoting Hybrid Rice Asexual Reproduction

Heterosis refers to the better performance of an F1 hybrid than both parent lines. However, this performance is easily lost due to the random segregation of genetic information in the offspring generations. Apomixis has the potential to allow offspring to retain valuable traits through asexual reproduction, which can lower seed production costs and is important for crop breeding, but apomixis is absent in major crops [122,123]. Thus, how to generate apomixis in crops has become a cutting-edge research hotspot in the field of botany. Recently, genome editing-mediated apomixis technology has realized heterosis fixation in hybrid offspring. In rice, clonal progeny retaining parental heterozygosity was obtained by CRISPR-editing *BABY BOOM1* (*BBM1*), *BBM2*, and *BBM3*, and its asexual-propagation traits can be stably inherited in multiple generations of clones [124]. Similarly, by combining heterozygosity fixation with haploid induction by simultaneous editing of *REC8*, *PAIR1*, *OSD1*, and *MTL*, Wang et al. generated hybrid rice plants that could propagate clonally through seeds, realizing self-propagation and stable transmission of elite F1 hybrid crops [125,126]. Excitingly, in hybrid rice, high-frequency synthetic apomixis was achieved by simultaneous editing of *PAIR1*, *REC8*, and *OSD1*, and clonal progeny could stably retain the phenotype and genotype of F1 hybrid in successive generations [127]. These studies suggest that the emergence of synthetic apomicts generated by CRISPR/Cas technology will provide efficient ways to utilize F1 hybrid heterosis, which will hopefully promote the realization of converting hybrids to apomixis in a sustainable way.

## 7. Facilitating Double Haploid Breeding Technology

Double haploid technology, including haploid induction and double haploid development, can greatly accelerate the breeding process by rapidly generating homozygous plants, and has been widely applied in various crops [5,128,129]. Using CRISPR/Cas genome editing technology, many advances have been made in the mechanisms and application of haploid induction in different crops [130]. In maize, key genes involved in haploid induction such as *ZmPOD65*, *ZmPLD3*, *ZmDMP7*, and *ZmMTL* have been characterized and show potential for breeding haploid inducers [131,132,133]. In rice, haploid induction was triggered by editing *OsMATL*, *OsECS1*, and *OsECS2*, respectively [125,134,135,136,137]. In *Brassica*, editing homologues of *DMP9* triggered haploid induction in *B. oleracea* and polyploid *B. napus*, offering haploid induction materials for efficient breeding [138,139]. In *Medicago truncatula*, haploid plants were generated by editing *DMP* homologues [140]. Moreover, editing *TaPLA*, *TaMTL*, and *TaCENH3α* could trigger haploid induction in wheat, indicating that CRISPR/Cas-mediated haploid induction could be extended from diploid crops to polyploid species [141,142,143,144]. These findings provide available methods for haploid induction in different crops.

Recently, CRISPR/Cas9 technology-mediated haploid induction systems have been developed in different crops. In maize, a haploid induction editing technology (HI-EDIT), a Haploid-Inducer Mediated Genome Editing (IMGE) system, an approach combining haploid induction with a robust haploid identification marker, and a CRISPR/dCas9-mediated gene activation toolkit were established to effectively generate genome-edited haploids [145,146,147,148]. Using a CRISPR/Cas9 vector with an enhanced green fluorescent protein expression cassette, an efficient haploid induction system was developed by editing *BnaDMP* genes in *Brassica napu* [149]. In foxtail millet, haploid induction has been achieved by CRISPR/Cas9-mediated mutation of *SiMTL*, providing a possible application of double haploid technology in its breeding [150]. Importantly, a fast technique for visual screening of wheat haploids was developed by combining the haploid inducer generated by editing *TaMTL* and embryo-specific anthocyanin markers, providing a promising strategy for a large-scale haploid inducer in different crops [151]. Recently, a RUBY reporter system, a background-independent and efficient marker for haploid identification, has been established, which enables easy and accurate haploid identification in maize and tomato, which will be promising in double haploid breeding in different crops [152].

## 8. Conclusions and Future Perspectives

The advent and updating of CRISPR/Cas technologies have paved the way for gene function analysis and crop breeding, providing unprecedented opportunities for the generation of novel genetic variation, rapid crop *de novo* domestication, creation of male-sterile lines and female-sterile lines, development of double haploid technology, and precise pyramiding breeding (Figure 1). In particular, the upgrade and integration of genome editing, haploid induction, and apomixis technologies will usher in a new era for crop breeding [125].

Although many CRISPR-Cas-edited crop materials have been generated in different crops, only few have been approved for commercial production and are entering the market worldwide due to regulatory policies such as a CRISPR/Cas9 waxy corn [26], a CRISPR-edited GABA-enriched tomato [153]. Currently, the United States, Japan, Brazil, Argentina, Israel, Canada, and Australia have adopted relatively loose regulatory policies on gene editing crops, that is, gene editing products that do not contain foreign genes are exempt from regulation. It is expected that more gene-edited plant products will accelerate commercialization. In 2021, the United Kingdom set to loosen rules for gene-edited crops and animals whose genes have been edited with precision techniques such as CRISPR. It will speed research and stimulate investment in these fields (https://www.science.org/content/article/uk-set-loosen-rules-gene-edited-crops-and-animals. Accessed on 26 May 2021). In China, the Guidelines for the Safety Evaluation of Gene Editing Plants for Agricultural Use (Trial) were issued in 2022, and the first safety certificate for the application of CRISPR-Cas-edited soybean was approved in 2023 (http://www.moa.gov.cn/ztzl/zjyqwgz/spxx/202304/t20230428_6426465.htm, accessed on 28 April 2023).

Notably, many CRISPR/Cas products have only been tested for their characters under simulated conditions, and there is a lack of field trials to evaluate their final field performance, which seriously hinders their application in production [100]. Thus, it is urgent to focus on field trials of CRISPR/Cas-edited crops and thus, promote their commercial production. For example, editing of *KRN2* or *OsKRN2* could significantly enhance maize and rice grain yield without apparent negative impacts on other agronomic traits in their field trials [20]. The *ZmRAVL1*-KO line displayed greater field yields than wild-type plants under different planting densities in two locations, showing excellent field traits for high planting density [22]. The higher yields and superior agronomies of 12 CRISPR/Cas waxy corn hybrids have been validated by field trials at 25 locations in the USA, and their precommercial production was launched in 2019 [26]. Thus, field trials like these will inevitably promote the commercial production of new gene-edited crops.

Low efficiency and high genotype dependency on genetic transformation processes are the major bottlenecks limiting the widespread application of CRISPR/Cas technology in different crops and elite varieties [154,155]. Recently, developed genotype-independent enhanced gene transformation systems, by overexpressing the morphogenic genes (*TaWOX5* and *Wus2/Bbm*), could significantly increase genome-edited plant regeneration in wheat, rye, barley, maize, and rice, providing new ways to expand genetic transformation and genome editing across the Poaceae family [155,156]. Further optimizing transformation methods will advance genome editing on a wider range of crop species and varieties. Moreover, it is imperative to develop CRISPR/Cas systems with higher editing efficiency, lower off-target activity, more editing ways, and wider editing range, which will make them more effective and flexible in crop breeding. Recently, an optimized Cas12a base editor (Cas12a-ABE) has been established to introduce inheritable multiplex base edits in wheat and maize, which will assist in optimizing genome editing systems in a wide range of crop species [157]. With the continuous development of CRISPR/Cas technology and its deep integration with other breeding techniques, it will become a popular strategy for breeders to precisely generate novel germplasms in different crops and usher in a new era of crop breeding.

## Figures and Tables

**Figure 1 plants-12-03119-f001:**
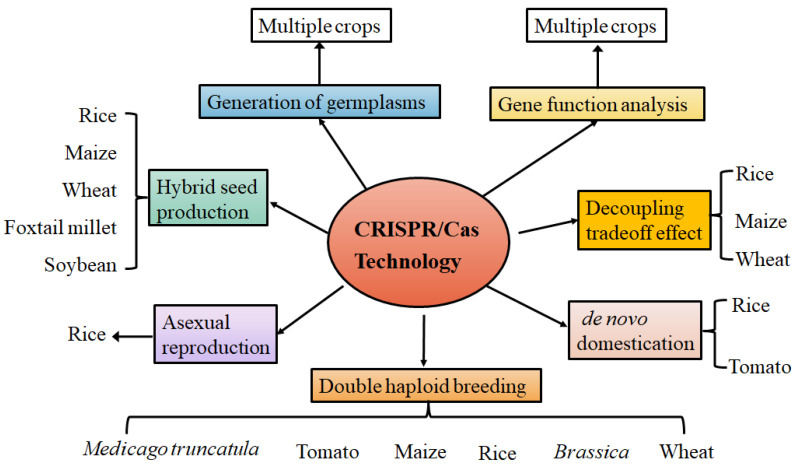
The application of CRISPR/Cas technology in the generation of new germplasms, analysis of gene function, and the iterative upgrading of multiple breeding technologies in different crops.

## Data Availability

All data have been included in the main text.

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
