# Peer review of "CRISPR/Cas Technology Revolutionizes Crop Breeding"

_plants, 2023, doi:10.3390/plants12173119_

Round 1

Reviewer 2 Report

This manuscript by Tang et al. conducted a thorough literature review focused on CRISPR uses for crop improvement in plant breeding areas. Their review significantly contributes to the scientific literature on using CRISPR/Cas-based tools in plant molecular breeding.

I suggest implementing a few corrections and improvements to enhance the quality of the manuscript. Here are my comments:

1.      The introduction is too short. Particularly, details about mutagenesis techniques for trait improvement in crop breeding will add value to the content and help readers. Authors should consider adding discussion with some relevant literature as below.

Plant mutagenesis tools encompass various techniques, including physical and chemical mutagens, PCR-based methods, T-DNA insertions, transposon insertions, RNA interference, and meganucleases (Reference- Evolution of plant mutagenesis tools: a shifting paradigm from random to targeted genome editing. https://doi.org/10.1007/s11816-019-00562-z)...

2.      Including a Figure describing the major research areas summarized in the MS (crop de novo domestication, decoupling the tradeoff effect, conventional hybrid seeds production, asexual reproduction, and double haploid breeding) would significantly enhance the manuscript quality.

3.      In introduction – Based on the initial CRISPR-Cas9, multiple developed CRISPR/Cas systems (including spCas9-NG, base editing, prime editing, xCas9, Cas12a /Cpf1, Cas12b, Cas13 and Cas14) have greatly improved editing effectiveness, precision and specificity, and enabled their application in gene knockout, base editing, prime editing, gene insertion, epigenetic modulation, transcriptional regulation, and RNA editing (Koonin et al. 2023).

It is a lengthy sentence with many details and needs to be revised with relevant literature. Here are suggestions-

Based on the initially engineered CRISPR/Cas9 (Jinek et al., 2012), diverse CRISPR tools have been developed for precise and effective genome editing across all layers of the central dogma processes (Pramanik et al., 2021). The expanding CRISPR toolbox enables gene knockout, gene deletion, base editing, gene insertion or replacement, targeted random mutagenesis, epigenetic modulation, transcriptional regulation, and RNA editing (Li et al., 2020; Shelake et al., 2022; Koonin et al. 2023).

Jinek et al., 2012 A programmable dual-RNA-guided DNA endonuclease in adaptive bacterial immunity. DOI: 10.1126/science.1225829

Li et al., 2020 Targeted, random mutagenesis of plant genes with dual cytosine and adenine base editors. https://doi.org/10.1038/s41587-019-0393-7

Pramanik et al., 2021 CRISPR-mediated engineering across the central dogma in plant biology for basic research and crop improvement. https://doi.org/10.1016/j.molp.2020.11.002

Shelake et al., 2022 Engineering drought and salinity tolerance traits in crops through CRISPR-mediated genome editing: Targets, tools, challenges, and perspectives.  https://doi.org/10.1016/j.xplc.2022.100417

4.      Please check the whole text for a uniform way of writing some basic terminologies, for example, CRISPR/Cas9 or CRISPR-Cas9.

5.      Also, check the whole text for writing gene names. It would be meaningful to italicize all gene names.

6.      Page 2- KRN2 gene name should be in italics.

7.      Page 6- Low efficiency and high genotype dependency are the major bottlenecks limiting widespread application of CRISPR/Cas technology in different crops and elite varieties (Altpeter et al. 2016; Wang et al. 2022a).

Consider this revised text for clarity –

Low efficiency and high genotype dependency on genetic transformation processes are the major bottlenecks limiting the widespread application of CRISPR/Cas technology in different crops and elite varieties (Altpeter et al. 2016; Wang et al. 2022a).

8.      Also, the following sentence on Page 6-

Recently, developed genotype-independent transformation systems by enhancing TaWOX5 and Wus2/Bbm could significantly increase transformation efficiency in wheat, rye, barley, maize and rice, providing new ways to expand genetic transformation and genome editing across the Poaceae (Wang et al. 2022a; Wang et al. 2023b).

Consider this revised text for clarity-

Recently, developed genotype-independent enhanced gene transformation systems by overexpressing the morphogenic genes (TaWOX5 and Wus2/Bbm) could significantly increase genome-edited plant regeneration in wheat, rye, barley, maize, and rice, providing new ways to expand genetic transformation and genome editing across the Poaceae family (Wang et al. 2022a; Wang et al. 2023b).

Moderate editing of English language required

Round 2

Reviewer 2 Report

The revised version is improved, and the authors address most comments. 

Regarding comment 2- Including a Figure describing the major research areas summarized in the MS (crop de novo domestication, decoupling the tradeoff effect, conventional hybrid seeds production, asexual reproduction, and double haploid breeding) would significantly enhance the manuscript quality.

The authors have added a Table and not the Figure. The editor may consider this aspect if it serves the same purpose of providing a better overview of reviewed literature. 

None.

Author Response

Dear Editor and Reviewers,

Thank your comments and suggesstions for our revised manuscript.

In this revision, we have provided a figure according your comments. Please you check and provide valuable suggestions.

Best regards

Haiwen
